# TOKEN BAYESIAN OPTIMIZATION: REASONING LLMS THINK BETTER WITH THE RIGHT LENGTH

## ABSTRACT

Reasoning-based Large Language Models (LLMs) exhibit strong capabilities in complex tasks such as mathematics, programming, and logic, with performance highly dependent on the length of the generated reasoning chains. However, the relationship between reasoning length and task performance is not simply linear; instead, it exhibits task-dependent, non-monotonic, and multi-peaked patterns. Short reasoning chains often result in incomplete arguments, while overly long ones may introduce noise or logical inconsistencies. Existing approaches such as reinforcement learning require extensive supervision or heuristic strategies based on fixed token budgets, and they struggle to effectively identify the optimal reasoning length. To address this, we propose Token Bayesian Optimization (TBO), a supervision-free and task-agnostic framework for reasoning length optimization. TBO combines coarse-grained boundary initialization with Bayesian iterative search, leveraging the evaluative power of LLMs to actively explore the token-length space and progressively converge toward the globally optimal reasoning point. Experiments on multiple standard reasoning benchmarks demonstrate that TBO consistently discovers reasoning lengths that better unlock the model's potential, achieving significant accuracy gains over existing baselines. The code is publicly available at: https://anonymous.4open.science/r/TBO-BEFD/.

## 1 INTRODUCTION

In recent years, large language models (LLMs) have demonstrated remarkable reasoning capabilities in complex tasks such as mathematical problem solving, code generation, and logical inference (Liu et al., 2025; Zhang et al., 2024a; Liang et al., 2025; Zhang et al., 2024b; Hong et al., 2024). These advances not only signal the growing maturity of LLMs in natural language understanding but also highlight their potential for tackling tasks that require deep and structured thinking. Researchers have explored efficient prompting strategies, such as Chain-of-Thought (CoT) and Direct Preference Optimization (DPO), to guide the models toward generating more structured and logically coherent reasoning. Building on this foundation, reasoning-oriented LLMs, such as Deepseek-R1 (DeepSeek-AI et al., 2025) and

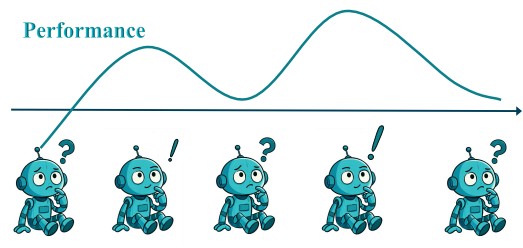

Figure 1: Schematic diagram of the relationship between reasoning length and performance: performance exhibits a multi-peak relationship with the reasoning length.

o3-mini (Ballon et al., 2025), have rapidly emerged as a key focus of contemporary research.

A critical insight revealed by recent studies (Han et al., 2024) is that there exists a task-dependent optimum relationship between reasoning length and model performance. When the reasoning chain is too short, the generative process lacks sufficient depth, leading to incomplete arguments and logical gaps (Chen et al., 2025). Conversely, excessively long chains introduce errors or irrelevant details that can contradict the original prompt, ultimately diminishing answer accuracy. This

non-monotonic relationship between reasoning length and performance presents a significant optimization challenge.

Current approaches to regulate reasoning depth fall into two main paradigms. The first uses continuous control based on reinforcement learning, exemplified by LCPO (Aggarwal & Welleck, 2025), which directs models to learn termination strategies through reward signals. However, these methods require extensive annotated datasets and develop task-specific policies that transfer poorly to new domains. The second approach employs heuristic strategies based on discrete thresholds, such as the Token Length Aware framework (Han et al., 2024) and TOPS (Yang et al., 2025b), which estimate and enforce token budgets during reasoning. While more lightweight, these methods provide only coarse-grained estimates and cannot accurately locate the optimal point on the reasoning length-performance curve. Further complicating this challenge, our analysis reveals that reasoning chains may exhibit multiple performance peaks as they increase in length, forming a complex, multiphase performance landscape. This raises a central challenge: **How can we effectively identify and guide the model towards the true global optimum among a set of unstable and task-dependent local peaks?**

Through our experiments, we found that for a given category of tasks, the underlying logical structure and required reasoning lengths tend to be fairly consistent, suggesting that identifying a single, globally optimal reasoning length for the entire task category is both feasible and effective. Based on this insight as well as former discussions, we propose a framework called **Token Bayesian Optimization (TBO)** which identifies the optimal reasoning length for a small number of representative examples and generalizes it across the entire task category, leveraging Bayesian optimization to model global uncertainty, enabling dynamic, task-adaptive exploration beyond unstable local peaks and guiding the search toward the true global optimum. TBO consists of the following three stages: **1) Searching Space Initialization:** We begin by selecting a set of key boundary points to construct an initial search space. Using an LLM-as-a-Judge posterior evaluation model, we assess performance across discrete reasoning lengths and identify an initial local optimum. While this point is likely not globally optimal, it serves as a strong starting point for further optimization. **2) Bayesian Iterative Exploration:** Building on the initial extrema, we apply Bayesian optimization to model and predict the reasoning performance over the entire token length space, guided by existing evaluation results. By continuously providing performance feedback through the LLM evaluator, the search space is updated and refined, gradually approaching the global optimum. **3) Convergence and Length Generalization:** Once the search space stabilizes, we retain the final identified optimum. By aggregating the results from multiple samples, we compute an average optimal length, which serves as the recommended reasoning token length for the entire task class. The main contributions of our work are:

1. We show that reasoning length optimization for a single task is not a simple single-peak search problem, but exhibits complex, non-monotonic patterns.

2. We propose Token Bayesian Optimization (TBO)—the first framework that achieves global reasoning length optimization without requiring labeled supervision or task-specific tuning.

3. We present extensive experiments demonstrating that TBO consistently improves accuracy across multiple benchmarks by adaptively allocating tokens. In addition, we show that reasoning length and performance exhibit a multi-peak relationship, that TBO remains robust under varying task difficulty distributions.

## 2 METHOD

Due to the non-monotonicity, non-continuity, and multi-peak of the relationship between reasoning chain length and reasoning performance, traditional gradient-based optimization methods and interval search iterative algorithms are not suitable.[1] Furthermore, using scalar metrics for evaluation not only introduces significant optimization cost but also makes it difficult to conduct a detailed analysis of the reasoning process. Therefore, we propose Token Bayesian Optimization (TBO), a method that progressively explores and converges toward the optimal reasoning length. The overall procedure is summarized in Algorithm 1, and the detailed algorithm of the GenerateNewTokens subroutine is provided in Appendix B.

---

[1]A detailed survey of related work is provided in Appendix A.

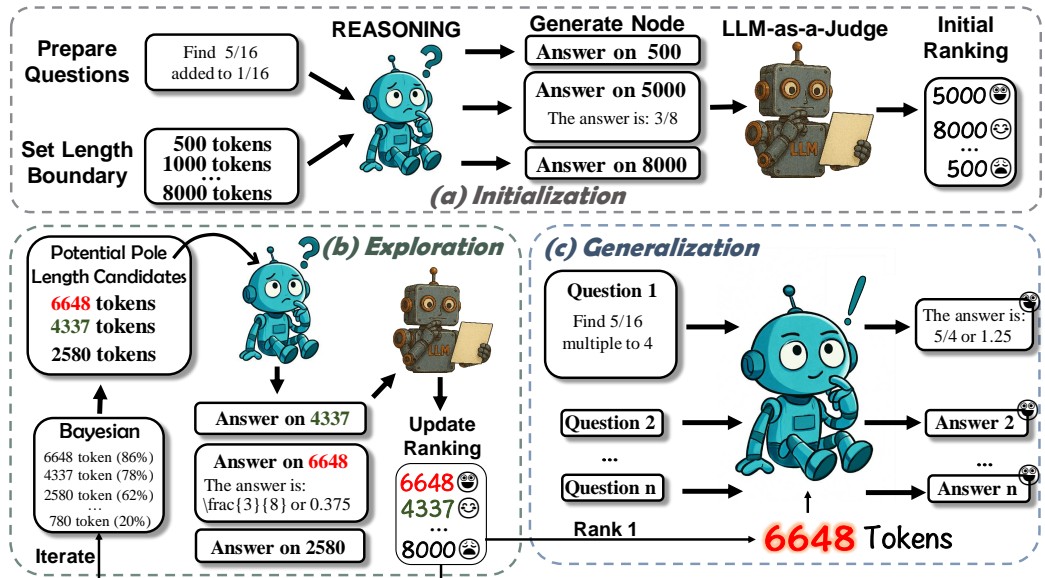

Figure 2: Overview of Token Bayesian Optimization (TBO). The process includes three main stages: (a) Initialization: boundary token lengths are selected and evaluated with an LLM judge to identify an initial optimum; (b) Exploration: Bayesian iterative optimization is performed, where a surrogate model is fit over token lengths, candidates are proposed and evaluated with the LLM; (c) Generalization: once the search stabilizes, the final optimum is selected and results are aggregated to recommend the optimal reasoning length for the task.

In this work, we implement optimization using Preferential Bayesian Search, with listwise ranking and LLM-as-a-judge for evaluating and acquiring preference signals. We construct a pipeline that consists of the following three keys. Through this pipeline, the TBO enables label-free, performance-driven control over the reasoning depth of LLMs, achieving adaptive enhancement of inference performance.

## 2.1 SEARCHING SPACE INITIALIZATION

We begin by defining a reasonable reasoning length range $\mathcal{L} = [100, 8000]$, and perform uniform sampling with a fixed step size $\Delta = 1000$ to construct the initial candidate set:

$$\mathcal{L}_{\text{init}} = \{l_1, l_2, \ldots, l_k\}, \quad l_i = 100 + (i-1) \cdot \Delta \tag{1}$$

For each candidate length $l_i \in \mathcal{L} * \text{init}$, we adopt a few-shot optimization strategy by sampling $1\%$ of similar[2] question instances $\mathcal{D} * \text{sub}$ from the dataset. For each instance, we generate a reasoning response $r\_i$ that includes both the reasoning process and final answer. Following the LCPO method(Aggarwal & Welleck, 2025), we insert a length control instruction into the original prompt using the following format:

$$prompt_{\text{n}} = \text{Concat}(prompt, \text{"Think for } l_n \text{ tokens."}) \tag{2}$$

We then cast the evaluation of reasoning lengths as a **Content-based Listwise Ranking** problem. For each $l_i$, the model produces an output $r_i$, and we prompt the LLM to globally rank the set $r_1, r_2, \ldots, r_k$ based on the overall quality of the responses. The resulting ranking $\mathcal{R} = \text{Rank}(r_1, \ldots, r_k)$ provides relative preference signals among the candidates. Unlike pairwise comparison methods, which only assess two candidates at a time, Listwise Ranking captures the global relationship among all candidates, enabling more accurate identification of performance trends. This is particularly beneficial in noisy settings, where local comparisons may be misleading, while global rankings help stabilize and guide the optimization trajectory.

---

[2]Specifically, we attempt multiple samples until we obtain a $1\%$ subset in which the token lengths differ by no more than $20\%$.

**Algorithm 1** TBO: Token Bayesian Optimization

---

**Require:** $T_0$: Initial token candidates, $K$: Max iterations, $\tau$: Early stop threshold
**Ensure:** $t^*$: Optimal token length
1: $T \leftarrow T_0, H \leftarrow T_0$             ▷ Current candidates and history
2: $A \leftarrow \{t : \texttt{CallForAnswers}(t) \mid t \in T\}$      ▷ Evaluate initial candidates
3: $T \leftarrow \texttt{Rank}(T, A)$                    ▷ Sort by performance
4: $t^* \leftarrow T[0], s \leftarrow 0$         ▷ Best token and stagnation counter
5: **for** $k = 1$ **to** $K$ **do**
6:      $T \leftarrow T[: \lfloor 2|T|/3 \rfloor]$             ▷ Keep top 2/3 candidates
7:      $T_{\text{new}} \leftarrow \texttt{GenerateNewTokens}(T, H, |T_0|) \setminus H$
8:      $A \leftarrow A \cup \{t : \texttt{CallForAnswers}(t) \mid t \in T_{\text{new}}\}$
9:      $T \leftarrow \texttt{Rank}(T \cup T_{\text{new}}, A)$
10:     $H \leftarrow H \cup T_{\text{new}}$
11:     $s \leftarrow (s+1) \cdot \mathbf{1}_{T[0]=t^*}$         ▷ Update stagnation counter
12:     $t^* \leftarrow T[0]$                     ▷ Update best token
13:     **if** $s \geq 3$ **or** $|T| < \tau$ **then**
14:         **break**
15:     **end if**
16: **end for**
17: **return** $t^*$

---

## 2.2 BAYESIAN ITERATIVE EXPLORATION

To efficiently optimize reasoning length in the absence of explicit scalar supervision, we adopt a **Preferential Bayesian Optimization (PBO)** framework that leverages pairwise preference feedback. This approach is well-suited for open-ended or black-box reasoning tasks where direct accuracy signals may be unavailable or unreliable. Let the discrete candidate reasoning lengths be denoted as $\mathcal{L} = \{l_1, l_2, \ldots, l_k\}$. Instead of relying on absolute performance scores, PBO models a latent utility function $f(l)$ using a Gaussian Process (GP), where preferences between length pairs $(l_i \succ l_j)$ are expressed probabilistically:

$$P(l_i \succ l_j) = \Phi\left(\frac{f(l_i) - f(l_j)}{\sqrt{2}\sigma}\right) \tag{3}$$

where $\Phi$ is the standard normal CDF and $\sigma$ is a noise parameter (Chu & Ghahramani, 2005). We place a Gaussian Process (GP) prior over $f(l)$, enabling uncertainty-aware modeling across the discrete candidate space.

Unlike standard preferential Bayesian optimization, where each iteration selects a new candidate point from a fixed space, our method operates directly on the reasoning token sequences. Specifically, in each iteration, we maintain a fixed sequence length and perform localized edits through a delete-and-replace strategy.

At each step $t$, given the current reasoning sequence of fixed length $L$. Based on previous evaluation signals, we identify and remove the bottom-performing one-third of these segments, yielding a truncated sequence $\tilde{S}_t$.

To compare the quality of the generated sequences, we prompt an external LLM-as-a-judge to rank them based on content relevance, coherence, and correctness (A discussion of the computational overhead of LLM-as-a-Judge is provided in Appendix C). This results in a listwise preference ordering, which is converted into multiple pairwise comparisons. Bayesian generates new nodes based on this preference relationship.

$$\mathcal{D}_t = \mathcal{D}_{S_t} \cup \{S_t^*\} \tag{4}$$

We model a latent utility function $f(S)$ over full sequences using a Gaussian process and adopt a preference-based acquisition function to select the most promising sequence $S_t^*$ for the next iteration. Note that our acquisition does not optimize over single token lengths, but over full sequence configurations derived from structured edits. Once the Bayesian optimization process converges, we retain the final optimum reasoning length. A theoretical justification of the convergence of our Bayesian optimization procedure is provided in Appendix D.

The surrogate model is updated based on the new posterior, and the process continues iteratively, refining the landscape of $f(l)$ and progressively converging toward the global optimum reasoning length $l_{\text{opt}}^*$. As high-performing regions emerge, the search space can be resampled with finer granularity to enhance resolution, while fallback strategies such as posterior mean sampling and random exploration ensure robustness against local optima.

## 3 EXPERIMENT

Table 1: Performance comparison on five benchmarks with o3-mini, o4-mini, and DeepSeek R1 . As CoT prompting is the default for all models, the CoT row denotes directly inputting the question without additional constraints. SPO is another prompting-based baseline, while TALE-EP is a length-control baseline. TBO is applied on top of CoT or SPO to optimize reasoning lengths.

| Base Model | Method | AGIEval-MATH | | GPQA-Diamond | | WSC | | BBH-Navigate | | StrategyQA | |
|---|---|---|---|---|---|---|---|---|---|---|---|
| | | ACC | Token | ACC | Token | ACC | Token | ACC | Token | ACC | Token |
| o3-mini | CoT | 0.6977 | 904 | 0.6939 | 3156 | 0.8782 | 643 | 0.996 | 849 | 0.7753 | 745 |
| | SPO | 0.6467 | 679 | 0.7500 | 2287 | 0.8635 | 591 | **1.0000** | 853 | 0.7753 | 571 |
| | TALE-EP | 0.6924 | 1193 | 0.7397 | 2616 | 0.8561 | 1047 | 0.9960 | 1437 | 0.7709 | 1239 |
| | CoT+TBO* | **0.7244** | 2113 | **0.7551** | 4019 | **0.8893** | 1596 | 0.9980 | 1613 | **0.7841** | 1725 |
| | SPO+TBO* | 0.6524 | 826 | **0.7551** | 2295 | 0.8819 | 648 | 0.9919 | 1019 | 0.7709 | 641 |
| o4-mini | CoT | 0.6486 | 1001 | 0.7614 | 2674 | **0.9449** | 362 | 0.9960 | 619 | 0.8202 | 533 |
| | SPO | 0.6334 | 1569 | 0.7424 | 2662 | 0.9121 | 889 | **1.0000** | 587 | 0.8166 | 1041 |
| | TALE-EP | 0.6460 | 1253 | 0.7626 | 2388 | 0.9158 | 806 | 0.9960 | 1020 | 0.8166 | 815 |
| | CoT+TBO* | **0.6486** | 1187 | 0.7727 | 2522 | 0.9341 | 745 | 0.9920 | 752 | **0.8238** | 781 |
| | SPO+TBO* | 0.6323 | 1550 | **0.9596** | 2796 | 0.9121 | 912 | 0.996 | 621 | 0.8166 | 815 |
| DeepSeek R1 | CoT | 0.8498 | 3432 | 0.7424 | 5601 | 0.8864 | 745 | 0.9280 | 1792 | 0.7118 | 638 |
| | SPO | 0.8130 | 2586 | - | - | 0.8571 | 884 | - | - | 0.6332 | 493 |
| | TALE-EP | 0.8693 | 2982 | 0.7172 | 6011 | 0.8959 | 973 | 0.9680 | 1703 | 0.7336 | 623 |
| | CoT+TBO* | 0.8444 | 3390 | **0.7828** | 5456 | **0.8974** | 827 | **0.9760** | 1668 | 0.7118 | 615 |
| | SPO+TBO* | **0.9889** | 1995 | - | - | 0.8864 | 835 | - | - | **0.7336** | 579 |

### 3.1 EXPERIMENT SETUP

To evaluate the performance of TBO, we conducted comprehensive experiments using three distinct large language models: DeepSeek R1  (DeepSeek-AI et al., 2025), o3-mini(Ballon et al., 2025) and o4-mini. Each model was systematically evaluated across multiple reasoning depths on a diverse benchmark suite of reasoning tasks. We used XML tags in prompts to structure outputs and employed GPT-3.5-turbo to validate and filter non-conforming responses.

Specifically, we used five challenging benchmarks: **1) AGIEval-MATH:** (Zhong et al., 2023) Containing 1,000 fill-in-the-blank math problems from high-level competitions such as AMC and AIME; **2) GPQA-Diamond:** (Rein et al., 2024) Comprising 198 expert-authored multiple-choice questions in biology, physics, and chemistry, specifically selected for their difficulty and objectivity; **3) WSC:** (Levesque et al., 2012) A commonsense reasoning dataset with 273 pronoun disambiguation tasks; **4) BBH-Navigate:** (Suzgun et al., 2023) A spatial reasoning task from BIG-Bench Hard that evaluates whether an agent returns to its starting point after following navigation instructions; **5) StrategyQA:** (Geva et al., 2021) Consisting of 2,780 yes/no questions requiring multi-step reasoning strategies.

We evaluated our TBO framework by integrating it with reasoning methods such as CoT and SPO, comparing these enhanced versions against their original baselines. We also included TALE-EP (Han et al., 2024), which uses expectation propagation to model token elasticity. Unlike TALE-EP, CoT (Wei et al., 2022) and SPO (Xiang et al., 2025) are reasoning strategies that TBO effectively augments.

For length optimization, we did not use the entire test set directly. Instead, we sampled a very small subset (about 1% of samples) to fit candidate lengths, and then applied the selected length to evaluate performance on the complete benchmark. This setup ensures that the optimization is data-efficient while reducing the risk of overfitting.

## 3.2 MAIN RESULT

As shown in Table 1, TBO demonstrates consistent performance across different datasets, consistently surpassing existing baselines. Notably, the optimal reasoning lengths discovered by TBO diverged significantly from those in prior naive settings, indicating substantial untapped optimization potential.

On the AGIEval-MATH benchmark, o3-mini with CoT+TBO leverages targeted exploration to identify a reasoning length that best aligns with accuracy, yielding a 3.8% accuracy improvement. In addition, o3-mini on GPQA-Diamond achieves an 8.8% relative accuracy gain with CoT+TBO, while DeepSeek R1 on StrategyQA shows a substantial 15.9% improvement with SPO+TBO. Furthermore, with o4-mini on GPQA-Diamond, SPO+TBO achieves a 29.2% accuracy increase over SPO alone, highlighting the strong synergy of the two methods. Similarly, for DeepSeek R1 on AGIEval-MATH, SPO+TBO discovers a reasoning length configuration that both reduces token usage by 22.8% and improves accuracy by 21.6%. These results suggest that existing naive settings often overlook the reasoning lengths most compatible with accuracy. By uncovering hidden potentials in underutilized model capacity, TBO effectively optimizes reasoning configurations while SPO expands the exploration space, collectively maximizing performance beyond existing naive limitations.

## 3.3 ABLATION ANALYSIS

We conducted an ablation experience on StrategyQA with o3-mini to isolate the impact of each component. Without Bayesian optimization, the optimal reasoning length was chosen only from a fixed set of discrete thresholds (e.g., from 100, 500, 1000 up to 12000 tokens), consistent with our original search space size. As for LLM-as-a-judge, we replaced the

Table 2: Ablation Study on StrategyQA using o3-mini. Eliminating the Bayesian search module and the Listwise ranking module respectively.

| o3-mini | ACC | Average Token |
|---|---|---|
| CoT+TBO | 0.7841 | 1725 |
| CoT+TBO w/o bayesian | 0.7527 | 1293 |
| CoT+TBO w/o LLM-Judge | 0.7493 | 1950 |

model-based evaluation with a simple ranking by exact match (or textual similarity) to the ground truth. All experiments were conducted under identical settings.

Table 2 clearly shows that both components are vital to TBO. When we remove Bayesian optimization and restrict the search to fixed thresholds, performance drops by roughly 4.0%. This not only reflects the loss of continuous, adaptive exploration, but also demonstrates that the true optimal reasoning length often does not coincide with commonly used discrete thresholds (e.g., 100, 500, 1000). Omitting the LLM-as-a-judge yields an even larger decline (4.3%), since exact-match or simple similarity to the ground truth cannot assess answer completeness, logical coherence, or necessary detail the way a dedicated LLM evaluator does. Specifically, Bayesian optimization uncovers fine-grained optima that lie between conventional checkpoints, while the evaluator provides the nuanced quality feedback needed to select the best candidate—losing either one degrades both accuracy and token efficiency.

## 3.4 FUTHER ANALYSIS

### 3.4.1 EVALUATOR CONSISTENCY AND BIAS ANALYSIS

To address potential bias or inconsistency when using large language models (LLMs) as evaluators, we designed a structured prompt with explicit ranking rules to minimize subjectivity. To achieve this, we conducted consistency experiments comparing evaluations from different LLMs (GPT-3.5-turbo, GPT-4o, GPT-o3) and human evaluators (three volunteers). Each evaluator selected the best option from a set of candidates generated during the optimization process, repeating the task three times with three candidate tokens. The results indicate that human evaluations exhibited more subjectivity and inconsistency. In contrast, LLM evaluations demonstrate higher stability and lower variance, suggesting that LLMs, within our framework, are less prone to bias than human evaluators. Detailed experimental results are provided in Appendix G.

### 3.4.2 MULTI-PEAK PATTERNS BETWEEN REASONING LENGTH AND LLM PERFORMANCE

The multi-peak relationship between reasoning length and model performance is the core motivation behind the design of TBO. To verify this phenomenon, we systematically varied the suggested reasoning length in fixed increments and recorded both the actual token consumption and the corresponding performance at each setting. These experiments were carried out on BBH-Navigate, using the locally deployed DeepSeek R1 model. By sweeping the suggested lengths from very short to well beyond typical requirements, we captured how actual token usage and accuracy co-vary, thereby testing the relationship between reasoning length and performance.

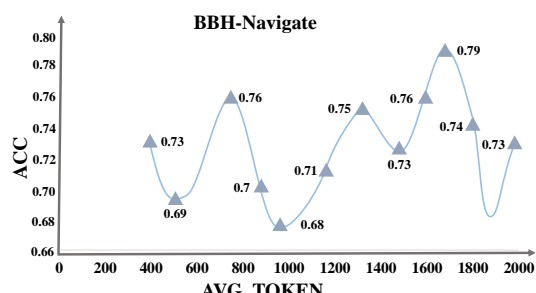

Figure 3: Schematic diagram of the relationship between reasoning model length and performance: the performance exhibits a multi-peak relationship with the reasoning length.

As shown in Figure 3, our experimental results in BBH-Navigate show that as the reasoning length increases, the precision of the answer does not follow a single-peak or monotonic pattern. Instead, multiple local maxima appear at different actual lengths. This multi-peak phenomenon is consistently observed across both datasets, providing strong evidence for our hypothesis that a multi-peak relationship exists between reasoning length and model performance. The theoretical justification for the emergence of such multi-peak patterns is also provided in Appendix E. We also analyzed the divergence between suggested and consumed lengths, with detailed results presented in Appendix F.

## 4 REASONING CHAIN LENGTH AND SOLUTION RELIABILITY – CASE STUDY

To further illustrate the relationship between reasoning length and answer reliability, we provide a detailed case study of a sequence reasoning problem. To investigate the impact of reasoning chain length on solution correctness, we attempted to solve the above problem using approaches with different reasoning depths. We applied a short reasoning chain, an optimal long reasoning chain selected by TBO, and an even longer chain, and compared the outcomes.

### 4.1 PROBLEM STATEMENT

**Question:** A strictly increasing sequence of positive integers $a_1$, $a_2$, $a_3$, ... has the property that for every positive integer $k$, the subsequence $a_{2k-1}$, $a_{2k}$, $a_{2k+1}$ is geometric and the subsequence $a_{2k}$, $a_{2k+1}$, $a_{2k+2}$ is arithmetic. Suppose that $a_{13} = 2016$. Find $a_1$.

### 4.2 VERY SHORT REASONING CHAIN

**Key reasoning steps:**

$$a_1 = x$$
$$a_{13} = r^{12}x$$
$$\text{Set } r = 2 \Rightarrow a_{13} = 4096x = 2016$$
$$\Rightarrow x = \frac{2016}{4096}$$

This approach makes the most naive substitution, assuming $a_{13}$ can be expressed as $a_1 \cdot f(r)$. Setting $r = 2$ directly gives a fractional solution for $a_1$, which is invalid. The chain is too short. It is quick to compute but ignores integer constraints entirely, leading to immediate failure.

### 4.3 SHORT REASONING CHAIN

**Key reasoning steps:**

$$a_1 = x$$
$$a_2 = rx, \quad a_3 = r^2 x$$
$$\dots$$
$$a_{13} = x \cdot [7r - 6]^2 = 2016$$
$$\text{Set } r = 6/5 \Rightarrow x = 350$$

Let $a_1 = x$ and use the problem's structure to derive $a_{13} = x \cdot [7r - 6]^2$. By setting $r = 6/5$, this method finds $a_1 = 350$. This short chain gives a seemingly reasonable integer solution 350, but it does not globally ensure integrality or uniqueness.

### 4.4 MEDIUM REASONING CHAIN

**Key reasoning steps:**

$$a_1 = 350, \quad a_2 = 420, \quad a_3 = 504$$
$$a_4 = a_3 + (a_3 - a_2) = 588$$
$$\text{Check whether } (a_2, a_3, a_4) \text{ forms a geometric sequence? No.}$$

Building on $a_1 = 350, r = 6/5$, this chain extends the reasoning to verify subsequent terms. It succeeds for the first few terms but breaks down at $a_4$, exposing inconsistency. This medium-length chain appears correct locally but reveals contradictions mid-way. The candidate $a_1 = 350$ is not globally valid.

### 4.5 OPTIMAL REASONING CHAIN (SELECTED BY TBO)

**Key reasoning steps:**

$$6r_1 - 5 = m \Rightarrow r_1 = \frac{m+5}{6}$$
$$a_1 = \frac{2016}{m^2}$$
$$\text{Enumerate } m = 2, 3, 4, 6, 12, \dots$$
$$\text{Check integrality step by step}$$
$$\text{Only } m = 2 \text{ works, yielding } a_1 = 504$$

A systematic approach introduces parameter $m$ with $6r_1 - 5 = m$, giving $a_1 = 2016/m^2$. Enumerating divisors of 2016 and checking integrality across the sequence yields the unique valid solution $a_1 = 504$. This chain length is optimal: long enough to verify all constraints, but not unnecessarily extended. It ensures both uniqueness and correctness.

### 4.6 LONG REASONING CHAIN

**Key reasoning steps:**

$$A_1 = a_1, \ B_1 = 2a_1$$
$$A_2 = 4a_1, \ B_2 = 6a_1$$
$$r_2 = \frac{B_2}{A_2} = \frac{3}{2}$$
$$A_3 = \frac{9}{4} \cdot 4a_1 = 9a_1$$
$$\dots$$
$$A_7 = P_7 a_1 = 2016 \Rightarrow a_1 = \frac{2016}{P_7}$$
$$\text{No verification that } P_7 \mid 2016$$

The chain is artificially extended by repeatedly introducing recurrences $r_k$, $A_k$, and $B_k$, under the assumption that any $r_1 > 1$ will work. Without checking integer constraints, it produces apparently self-consistent but actually invalid results. This long reasoning chain accumulates unchecked assumptions. Although it looks coherent, it generates invalid or non-integer results, showing that "longer" does not always mean "better."

Table 3: Difficulty distribution analysis of GPQA-Diamond, StrategyQA, and WSC benchmarks evaluated using o3-mini and DeepSeek R1 models.

| Base Model | Dataset | Count | Mean | Std Dev | Min | Max |
|---|---|---|---|---|---|---|
| o3-mini | StrategyQA | 227 | 908.12 | 614.85 | 215 | 3808 |
| | GPQA | 198 | 3290.53 | 2860.30 | 391 | 14787 |
| | WSC | 270 | 756.19 | 460.20 | 266 | 2605 |
| DeepSeek R1 | StrategyQA | 229 | 506.59 | 387.61 | 36 | 1475 |
| | GPQA | 198 | 5327.11 | 2713.06 | 381 | 12136 |
| | WSC | 273 | 902.75 | 668.26 | 278 | 3930 |

## 4.7 ROBUSTNESS TO DATASET DIFFICULTY VARIANCE

The core theoretical foundation of TBO's generalization ability lies in the assumption that most instances within a task can benefit from a shared, globally optimal reasoning length. When reasoning complexity is uniform, one optimized length generalizes well and yields stable gains. When complexity varies widely, a fixed length cannot fit all cases, reducing overall improvement.

To test this hypothesis, we conducted a cross-task analysis on three representative reasoning benchmarks—GPQA, StrategyQA, and WSC. We compared token usage characteristics across datasets, including statistics such as the mean and variance, and aligned these with the performance gains achieved through TBO. Table 3 shows that GPQA has the highest standard deviation in actual token usage, which is expected since its 198 multiple-choice questions cover a wide range of topics in biology, physics, and chemistry. This leads to large differences in the optimal reasoning length needed for each question. StrategyQA shows moderate variation, while WSC is the most consistent, which aligns with TBO achieving its best optimization results on WSC. Overall, these findings suggest that TBO works best when question difficulty is relatively uniform, as a single reasoning length can suit most examples. In contrast, when difficulty varies widely, it becomes harder to find one reasoning length that fits all, and the optimization gains are reduced. Then a clear pattern emerges: the difficulty of a question is linearly associated with the reasoning length required by the model, with more difficult problems demanding longer reasoning chains. A more detailed empirical analysis of token length distributions across datasets is provided in Appendix H.

## 5 CONCLUSION

We focus on a central challenge in large language model reasoning: how to determine the optimal reasoning length to maximize task performance. The relationship between reasoning depth and accuracy exhibits a multi-peaked landscape—several lengths may yield reasonable results, but only specific ones achieve global optimality. Existing methods, which largely rely on fixed or heuristic length settings, have long overlooked this critical factor, thereby constraining model potential. To address this, we propose Token Bayesian Optimization (TBO), a lightweight, model-agnostic framework that leverages iterative LLM feedback to efficiently identify globally optimal reasoning lengths without labeled supervision or reinforcement learning.

Across diverse benchmarks, TBO improves performance across models and tasks, adapting reasoning length to task complexity. These demonstrate not only the effectiveness of reasoning length optimization but also the substantial untapped potential within current prompting paradigms. In future work, we aim to develop finer-grained optimization that adapts reasoning length to variations in task type and difficulty, further enhancing robustness in heterogeneous settings.

## 6 REPRODUCIBILITY STATEMENT

We have made every effort to ensure the reproducibility of our work. An anonymous link to the source code is provided in the supplementary materials, enabling others to replicate our implementation. In addition, the appendix contains clear explanations of all underlying assumptions as well as complete proofs of the theoretical results. For the empirical studies, we also release the datasets used in our experiments together with detailed data processing steps. These resources collectively support the faithful reproduction and verification of our findings.

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
