APPENDIX

# A  RELATED WORK

## A.1  PROMPTING FOR REASONING CONTROL

Prompt engineering is crucial for enhancing LLM reasoning, with strategies like Chain-of-Thought (CoT) (Wei et al., 2022), Self-Consistency (Wang et al., 2023b), and Least-to-Most Prompting (Zhou et al., 2023a) guiding intermediate reasoning steps. However, these often incur high token costs and lack fine control, limiting use in constrained settings. To address this, automatic prompt optimization methods leveraging LLM-as-a-Judge, such as PromptAgent (Wang et al., 2024), Prompt-Breeder (Fernando et al., 2023), and TextGrad (Yüksekgönül et al., 2024), improve prompts but usually depend on ground truth.

Self-supervised Prompt Optimization (SPO) (Xiang et al., 2025) has been proposed to minimize the reliance on human supervision by automatically generating preference signals from model outputs. This approach integrates the LLM-as-a-Judge paradigm (Zheng et al., 2023) with pairwise comparison frameworks (Liu et al., 2024) to enable automated evaluation and iterative refinement of prompts, thereby facilitating efficient and scalable prompt optimization without extensive human annotation. Further efforts, such as those by ProTeGi (Pryzant et al., 2023) and Human-Level Prompt Engineers (Zhou et al., 2023b), explore automatic prompt selection and fine-tuning. GLaPE (Zhang et al., 2024c) advances this line of work by proposing a label-agnostic output scoring mechanism, pushing prompt optimization toward greater efficiency and lower cost.

## A.2  REASONING LENGTH CONTROL AND COMPRESSION

Chain-of-Thought (CoT) prompting enhances the reasoning abilities of language models on complex tasks. However, studies have shown that excessively long reasoning paths may actually degrade performance—an effect that is particularly pronounced in smaller models (Yang et al., 2025a). Moreover, long input sequences tend to impair reasoning accuracy, revealing a non-monotonic relationship between token count and output quality (Wu et al., 2025; Levy et al., 2024).

To mitigate performance degradation from excessively long reasoning paths, methods for controlling reasoning length and reducing token usage have been proposed. LCPO[2] uses reinforcement learning to optimize token usage within a fixed budget (Aggarwal & Welleck, 2025), while the budget-forcing strategy adjusts reasoning time during testing for performance tuning (Muennighoff et al., 2025). Compression strategies, such as the shortest effective reasoning for simple tasks (Zhang et al., 2025) and token budget-based compression (Han et al., 2024), aim to reduce reasoning costs. TokenSkip (Xia et al., 2025) prunes less important tokens for compressed reasoning, and (Chen et al., 2025) suggests dynamically adjusting reasoning length based on task complexity.

## A.3  BAYESIAN OPTIMIZATION FOR STRUCTURE SEARCH

Bayesian Optimization (BO) is a principled, sample-efficient method for expensive structure search tasks—such as density functional theory simulations or molecular synthesis—by modeling the objective with a surrogate and guiding sampling via acquisition functions (Frazier, 2018; Snoek et al., 2012; Shahriari et al., 2015). Unlike random or heuristic searches, BO balances exploration and exploitation by quantifying both predicted performance and uncertainty (Jones et al., 1998), enabling effective optimization in hyperparameter tuning (Oliver & Wang, 2024), model fusion (Jang et al., 2024), and inference configuration (Wang et al., 2023a) under diverse budgets and model scales.

The acquisition function in BO is essential for guiding the optimization process. It balances exploration of unknown regions and exploitation of areas with known high performance. By evaluating this trade-off, the acquisition function directs the search towards promising solutions while ensuring that less explored areas are also considered. For our approach, we use EUBO (Expected Utility of Bayesian Optimization) as the acquisition function. EUBO estimates the expected utility of each

---

[2]We do not report LCPO as a baseline, since its objective is to enforce short reasoning chains within a fixed budget via reinforcement learning, whereas TBO aims to explore and identify the performance-optimal length. The two methods target fundamentally different goals.

candidate solution by factoring in both predicted performance and associated uncertainty. This allows the optimization to efficiently explore the search space and move toward the optimal solution.

Notably, BO extends beyond training to prompt and instruction optimization in black-box language models. Approaches like BOInG (Sabbatella et al., 2024) and HbBoPs (Schneider et al., 2024) leverage BO to navigate large combinatorial prompt spaces by embedding prompts into structured or continuous representations.

Preferential Bayesian Optimization (PBO) further advances this by using pairwise or ranking-based feedback instead of explicit objective values, making it especially suitable when direct evaluations are noisy, costly, or subjective (González et al., 2017; Chu & Ghahramani, 2005).

## B    DETAILED ALGORITHM FOR GENERATENEWTOKENS

---

**Algorithm 2** GenerateNewTokens

---

**Require:** $T$: Current token ranking, $H$: Token history, $n$: Target candidate count
**Ensure:** $T_{new}$: Novel token candidates
1: $T_{new} \leftarrow \emptyset$
2: **while** $|T| + |T_{new}| < n$ **do**
3:     $t \leftarrow \texttt{SuggestNext}(T)$                              ▷ Bayesian proposal
4:     **if** $t \notin H$ **then**
5:         $T_{new} \leftarrow T_{new} \cup \{t\}$
6:     **end if**
7: **end while**
8: **return** $T_{new}$

---

## C    COMPUTATIONAL OVERHEAD OF LLM-AS-A-JUDGE

**Iteration rounds and judge calls.**    Our iterative optimization typically converges within about 10 rounds. The initial candidate sequence length is usually set to 9. At each round, we remove the bottom third of candidates (by the current listwise ranking) and add 3 new ones to maintain the sequence length. If a newly proposed candidate has appeared before or is filtered as invalid, it is skipped, which may temporarily shorten the sequence. Each round requires only one call to the LLM-as-a-Judge for listwise ranking, so the maximum number of judge calls is approximately 82 (i.e., $10 + (9 + 39) \times 2$) under our settings.

**Observed token consumption during optimization (o3-mini).**    The table below reports the total tokens consumed by the LLM-as-a-Judge during the optimization phase, when TBO is applied on top of CoT or SPO. These costs are separate from the final inference cost.

Table 4: Token consumption of the LLM-as-a-Judge during TBO optimization (o3-mini).

| Task | CoT+TBO | SPO+TBO |
|------|---------|---------|
| AGIEval-MATH | 211522 | 196225 |
| GPQA | 181057 | 165956 |
| WSC | 267891 | 277206 |
| BBH-Navigate | 142003 | 96442 |
| StrategyQA | 380608 | 270713 |

## D    THEORETICAL JUSTIFICATION OF BAYESIAN OPTIMIZATION CONVERGENCE

Bayesian Optimization (BO) is widely used for optimizing expensive black-box functions. This section provides a theoretical justification that BO can converge to the global optimum under certain assumptions, using well-established results from the literature.

PROBLEM SETUP

We aim to optimize an unknown function $f : \mathcal{X} \to \mathbb{R}$ over a compact domain $\mathcal{X} \subset \mathbb{R}^d$. The global optimum is denoted as:

$$x^* = \arg\max_{x \in \mathcal{X}} f(x)$$

Bayesian Optimization constructs a probabilistic surrogate model $f^\dagger$ (typically a Gaussian Process) and selects query points according to an acquisition function.

GAUSSIAN PROCESS PRIOR

We assume $f$ is modeled as a Gaussian Process (GP):

$$f(x) \sim GP(m(x), k(x, x'))$$

where $m(x)$ is the prior mean (often set to 0), and $k(x, x')$ is the covariance kernel (e.g., RBF or Matérn kernel). After $t$ observations $\mathcal{D}_t = \{(x_i, f(x_i))\}_{i=1}^t$, the posterior distribution is:

$$f(x) \mid \mathcal{D}_t \sim \mathcal{N}(\mu_t(x), \sigma_t^2(x))$$

ACQUISITION FUNCTION

The acquisition function $\alpha_t(x)$ balances exploration and exploitation. Common choices include: - Upper Confidence Bound (UCB):

$$\alpha_t(x) = \mu_t(x) + \sqrt{\beta_t}\sigma_t(x)$$

- Expected Improvement (EI):

$$\alpha_t(x) = \mathbb{E}[\max(0, f(x) - f(x^+))] = (\mu_t(x) - f(x^+))\Phi(z) + \sigma_t(x)\phi(z),$$

where $z = \frac{\mu_t(x) - f(x^+)}{\sigma_t(x)}$, and $\Phi, \phi$ denote the CDF and PDF of the standard normal distribution.

THEORETICAL GUARANTEE

The GP-UCB convergence theorem (Srinivas et al., 2009) states that, under mild conditions, the sequence of points

$$x_{t+1} = \arg\max_{x \in \mathcal{X}} \mu_t(x) + \sqrt{\beta_t}\sigma_t(x)$$

satisfies, with high probability:

$$f(x^*) - f(x_t) \leq O\left(\sqrt{\frac{\beta_t \gamma_t}{t}}\right)$$

where $\beta_t = 2\log\left(\frac{|\mathcal{X}|\pi^2 t^2}{6\delta}\right)$ and $\gamma_t = \max_{A \subset \mathcal{X}, |A|=t} I(y_A; f)$. Therefore,

$$\lim_{t \to \infty} f(x_t) \to f(x^*)$$

CUMULATIVE REGRET BOUND

The cumulative regret is defined as:

$$R_T = \sum_{t=1}^T \left[f(x^*) - f(x_t)\right]$$

It follows that:

$$R_T = O\left(\sqrt{T\beta_T\gamma_T}\right), \quad \text{hence} \quad \frac{R_T}{T} \to 0 \quad \text{as } T \to \infty$$

INFORMATION GAIN

For different kernels, the information gain $\gamma_T$ can be bounded as: - RBF kernel:

$$\gamma_T = O\big((\log T)^{d+1}\big)$$

- Matérn kernel:

$$\gamma_T = O\Big(T^{\frac{d(d+1)}{2\nu+d(d+1)}} \log T\Big)$$

These bounds ensure efficient exploration even in multimodal settings.

## E  PROOF OF MULTI-PEAK RELATIONSHIP BETWEEN REASONING LENGTH AND PERFORMANCE

### E.1  PROBLEM SETTING.

Let $L > 0$ denote the reasoning length (e.g., number of tokens). We consider three types of information in the reasoning process:

- **Effective information** $I_{\text{eff}}(L)$: Directly improves performance, saturates as $L$ increases;
- **Ineffective (noisy) information** $I_{\text{ineff}}(L)$: Accumulates with $L$ and impairs performance;
- **Potentially effective information** $I_{\text{pot}+}(L)$: Initially ineffective, but can be activated under certain conditions to enhance performance.

Let the final performance metric (e.g., accuracy or utility) be $P(L)$, a function of these three components.

### E.2  STEP 1: MATHEMATICAL MODELING OF VARIABLES

The three types of information are modeled as:

$$I_{\text{eff}}(L) = I_{\max}\left(1 - e^{-\kappa L}\right), \quad I_{\max} > 0, \kappa > 0$$

$$I_{\text{ineff}}(L) = \eta L^{\alpha}, \quad \eta > 0, \alpha \geq 1$$

$$I_{\text{pot}}(L) = \xi f(L)$$

Here, $f(L)$ may be a periodic or oscillatory function (e.g., $f(L) = \sin(\omega L + \phi)$), reflecting that certain hidden information is only activated at specific reasoning lengths.

### E.3  STEP 2: PERFORMANCE FUNCTION AND BOUNDEDNESS

We define the performance function as

$$P(L) = \sigma\Big(\beta_1 I_{\text{eff}}(L) + \beta_2 h\big(I_{\text{pot}}(L)\big) - \gamma I_{\text{ineff}}(L) + b\Big)$$

where

- $\sigma(x)$ is a bounded activation function such as the sigmoid, ensuring $P(L)$ is always bounded.
- $h(\cdot)$ is an activation or gating function for the potential information, for example, $h(x) = \max\{0, x - \theta\}$
- $\beta_1, \beta_2, \gamma, b$ are real coefficients (weights and bias).

Therefore, for all $L > 0$, $P(L)$ is bounded.

### E.4   STEP 3: EXISTENCE OF EXTREMA AND MULTI-PEAK PROPERTY

Consider the derivative with respect to $L$:

$$\frac{dP}{dL} = P(L)\big(1 - P(L)\big) \cdot G(L)$$

where

$$G(L) = \beta_1 I_{\max}\kappa e^{-\kappa L} + \beta_2 h'\big(I_{\mathrm{pot}}(L)\big)I'_{\mathrm{pot}}(L) - \gamma\eta\alpha L^{\alpha-1}$$

The first term is positive and decreases monotonically. The third term is negative and grows in magnitude with $L$. The second term, involving $I_{\mathrm{pot}}(L)$, can oscillate if $f(L)$ is periodic or has non-monotonic activations. As a result, $G(L)$ may change sign multiple times, causing $\frac{dP}{dL}$ to have multiple roots and thus multiple local maxima and minima of $P(L)$.

### E.5   STEP 4: EXISTENCE OF OPTIMAL VALUES

Since $P(L)$ is continuous and bounded, it must attain its supremum for some $L_k^* > 0$:

$$\max_{L>0} P(L) = P(L_k^*)$$

where $L_k^*$ may not be unique (multiple local maxima may exist).

The relationship between reasoning length and final performance is bounded, nonlinear, and exhibits multiple peaks. Performance is jointly determined by effective information, noise accumulation, and the activation of potentially effective information at certain reasoning lengths. Due to continuity and boundedness, one or more optimal values always exist.

## F   DIVERGENCE BETWEEN SUGGESTED AND CONSUMED LENGTHS

The suggested reasoning length often deviates from the model's actual reasoning length, and the relationship between the two is far from linear. Prior work such as (Han et al., 2024) has identified a phenomenon known as Token Elasticity. Specifically, as we gradually increase the suggested reasoning length, the model's actual token consumption does not grow linearly. Instead, once the suggestion exceeds a certain reasonable range, actual usage begins to decline. This phenomenon highlights that while we suggest an optimal token length for reasoning, the model's actual output token count may vary, but typically stabilizes around an effective range that maximizes performance.

To investigate this effect, we conducted a systematic experiment using the o3-mini model on the GPQA and StrategyQA datasets. We swept the suggested reasoning length from 0 to 12,000 tokens in increments of 100, recording the model's actual token consumption at each step to map the global relationship between the suggested and consumed tokens. While the model may not strictly adhere to the suggested length, our results show that the actual token consumption tends to stabilize near an optimal length after a few iterations, which significantly improves reasoning performance without the need for strict token length enforcement.

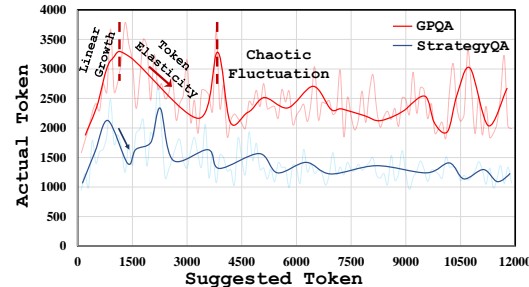

Figure 4: Token consumption in reasoning tasks shows three distinct phases: linear growth, elastic plateau, and chaotic fluctuation.

Our experimental results show a consistent three-phase pattern across both datasets. In the first phase, when the suggested length is below the minimal requirement to generate an answer, actual consumption grows approximately linearly with the suggestion—indicating that the suggested length serves as a useful control signal. In the second phase, once the suggestion enters a reasonable range, the Token Elasticity effect emerges: further increases in suggestion can lead to reduced actual consumption. Beyond a certain threshold,

the relationship between suggested and actual reasoning length enters a third phase characterized by increasing nonlinearity and chaotic behavior, indicating a more complex underlying interaction.

Interestingly, this phenomenon reveals an important insight: within the early-stage ideal interval, it is possible to reach a local performance peak by traversing the suggested length space. However, once beyond this ideal region, the relationship between the suggested and actual reasoning lengths becomes increasingly complex and unpredictable, making it difficult to capture with any regular or rule-based method. This highlights a key advantage of TBO: rather than relying on a linear assumption, it leverages Bayesian optimization to model global uncertainty and identify the most promising extrema across the entire space.

## G DETAILED RESULTS ON EVALUATOR CONSISTENCY

Table 5: Consistency of LLM and Human Evaluation Methods. The values in curly braces represent the reasoning lengths (in tokens) selected by each evaluator over three rounds.

| Evaluation Method | GPT-3.5-turbo / Person 1 | GPT-4o / Person 2 | GPT-3 / Person 3 |
|---|---|---|---|
| LLM | {5470, 3612, 5470} | {5470, 5470, 5470} | {5470, 5470, 5470} |
| Human Evaluation | {3612, 3612, 3053} | {3612, 5470, 5470} | {9919, 9919, 5470} |

The table presents the evaluation consistency between LLMs and human evaluators across three models (GPT-3.5-turbo, GPT-4o, and GPT-3). For example, in the GPT-3.5-turbo case, LLM evaluations consistently selected 5470 tokens, while human evaluators showed more variation, with selections ranging from 3053 to 3612 tokens. This highlights the higher stability of LLM evaluations compared to human evaluations, demonstrating that LLMs are less prone to inconsistencies and bias within our designed framework.

## H ROBUSTNESS TO DATASET DIFFICULTY VARIANCE

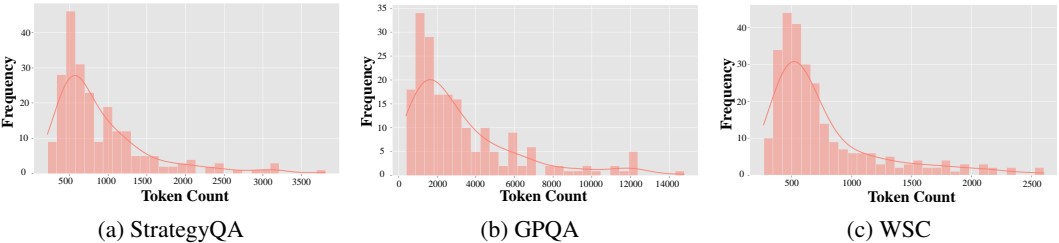

(a) StrategyQA        (b) GPQA        (c) WSC

Figure 5: Token Consumption Distribution for StrategyQA, GPQA, and WSC.

The token count frequency histograms for StrategyQA, GPQA, and WSC (Figures 5a,5b,5c) reveal a pronounced clustering in the reasoning lengths required by individual questions within each dataset. Specifically, for WSC, the histogram shows a highly concentrated distribution, indicating that the majority of questions require a similar number of tokens for optimal reasoning. GPQA and StrategyQA also display clear peaks, though GPQA exhibits a broader spread, consistent with its higher standard deviation reported in Table 3.

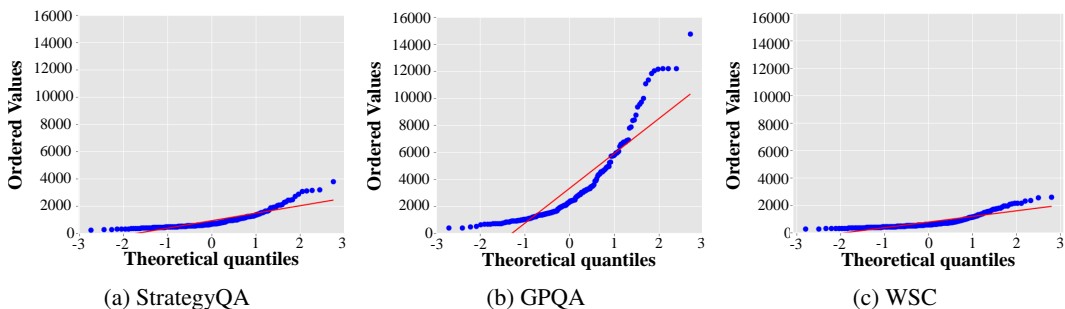

Figure 6: Quantile–Quantile Plots for StrategyQA, GPQA, and WSC.

The accompanying Quantile–Quantile Plots (QQ-plots) (Figures 6a,6b,6c) further confirm these observations. For WSC, the QQ-plot closely aligns with the reference line, suggesting that token usage across questions is not only concentrated but also follows a relatively normal distribution. In contrast, GPQA's QQ-plot deviates more significantly from the diagonal, indicating heavier tails and greater heterogeneity in token requirements across questions.

These empirical findings support our core hypothesis: TBO delivers the most substantial and stable performance improvements in scenarios where the token consumption per question is tightly clustered. In such cases, a globally optimized reasoning length suffices for the majority of instances, maximizing both efficiency and accuracy. However, when the token usage distribution is more dispersed—as in GPQA—the benefits of a single optimized length are diminished, reinforcing the need for more granular, instance-level control over reasoning length. This result underscores the critical role of within-task difficulty distribution in shaping the effectiveness boundaries of length optimization strategies such as TBO.

## I USE OF LARGE LANGUAGE MODELS

We confirm that no large language models were used in the process of research ideation, experimentation, analysis, or writing of this paper. All contributions, including conceptual development, implementation, and manuscript preparation, were carried out entirely by the authors without the assistance of LLM-based tools.