# OpenReview forum: "Token Bayesian Optimization: Reasoning LLMs Think Better with the Right Length"
_ICLR.cc/2026/Conference — Submitted to ICLR 2026_

### Official Review · Reviewer_J2BH · 2025-10-31

**Soundness:** 1
**Presentation:** 2
**Contribution:** 1
**Rating:** 2
**Confidence:** 3

**Summary:**

This paper proposes Token Bayesian Optimization (TBO), a framework for optimizing the reasoning length of large language models. The authors argue that the relationship between reasoning chain length and task performance is non-monotonic and often exhibits multiple peaks. To address this, TBO combines boundary-based initialization, Bayesian iterative search, and LLM-as-a-judge evaluation to identify an approximately optimal reasoning length for each task category. The framework's effectiveness is demonstrated through experiments on benchmarks such as AGIEval-MATH, GPQA-Diamond, WSC, BBH-Navigate, and StrategyQA with models including o3-mini, o4-mini, and DeepSeek R1. Results show that applying TBO on top of baseline prompting strategies yields higher accuracy, often with increased token usage. Taken together, these findings position TBO as a supervision-free, task-agnostic method for length optimization in reasoning LLMs.

**Strengths:**

The paper highlights the non-monotonic relationship between reasoning length and performance and attempts to formalize this problem as an optimization task.

**Weaknesses:**

1. The assumption of a single global optimal length per task may be overly simplistic, since problems of varying difficulty could require different reasoning lengths, as suggested by the variance reported in Table 3.
2. The approach is described as building on LCPO by controlling reasoning length via prompt instructions. However, LCPO notes that models do not always reliably follow length constraints in prompts.
3. The method description lacks sufficient detail for full reproducibility. Key specifics and hyperparameters are missing. Furthermore, the motivation for choosing Preferential Bayesian Search over other black-box optimization methods is not thoroughly discussed. A justification for why this preference-based approach is particularly suited for this problem would strengthen the method section.
4. Most of the reported performance gains (Table 1) come at the cost of higher token usage, making it unclear whether benefits stem from TBO or simply longer reasoning.
5. There is a discrepancy between Figure 3 and Table 1 regarding DeepSeek R1 performance on BBH-Navigate. Figure 3 shows performance peaks between 0.68–0.79, whereas Table 1 reports much higher values (0.92–0.97).

**Questions:**

1. Could you please provide more details on the Preferential Bayesian Optimization？ What was the rationale behind these specific choices?

2. How sensitive is TBO to the exact wording of the length control instruction? Furthermore, could you provide an ablation study on the size of the subset used for optimization (e.g., 0.5%, 1%, 2%) to demonstrate the robustness of this choice? What impact do these factors have on the efficiency and effectiveness of the approach?

3. How well do the LLMs actually comply with the specified token length budget？ Could you report statistics on the actual generated token lengths versus the target length L across your experiments (e.g., mean absolute error, standard deviation)?

4. The LLM judge's list-wise ranking is central to TBO. Beyond the provided case study and consistency check, could you perform a quantitative analysis to validate its accuracy?

---

### Official Review · Reviewer_U1aY · 2025-10-31

**Soundness:** 2
**Presentation:** 1
**Contribution:** 2
**Rating:** 4
**Confidence:** 3

**Summary:**

The paper addresses the non-monotonic relationship between reasoning length and model performance in LLMs. While short reasoning chains are often incomplete, overly long ones introduce noise and contradictions. To find the optimal reasoning length, the authors propose Token Bayesian Optimization (TBO), a supervision-free, model-agnostic framework that employs Bayesian optimization guided by an LLM-as-a-judge to explore reasoning-length space and converge on a global optimum.

Experiments on five reasoning benchmarks with three LLMs (o3-mini, o4-mini, DeepSeek-R1) show that TBO improves accuracy by 3–29% relative to CoT and other length-control baselines such as TALE-EP. Ablations confirm that both Bayesian search and LLM-as-a-Judge are necessary for full performance.

**Strengths:**

1. The formulation and motivation of this paper are interesting.

2. TBO is tested across multiple models and diverse reasoning tasks, showing improvements, supporting generality.

**Weaknesses:**

1. The related work is deferred to the appendix rather than integrated into the main paper. Including it in the main text would provide essential context, clarify positioning relative to prior methods, and make the work more self-contained and complete.

2. TBO treats reasoning length as a black-box parameter; there is little explanation or modeling of why certain lengths work better, or how they interact with internal reasoning dynamics. In other words, I think length should not be an dependant variable of model performance, but an outcome of reasoning dynamics.

3. It would be better to add computation and overhead analysis for efficiency and scalability.

**Questions:**

See above.

---

### Official Review · Reviewer_ePjc · 2025-11-02

**Soundness:** 3
**Presentation:** 2
**Contribution:** 2
**Rating:** 4
**Confidence:** 4

**Summary:**

The paper proposes Token Bayesian Optimization (TBO), a supervision-free, task-agnostic framework to pick an optimal reasoning token length for LLMs. The key claim is that accuracy vs. reasoning length is non-monotonic and multi-peaked, so fixed budgets or simple heuristics underperform. TBO: (i) initializes a search over boundary lengths, (ii) performs preferential Bayesian optimization with a GP surrogate using LLM-as-a-judge listwise rankings as feedback, and (iii) converges to a task-level recommended length which is then applied to the full benchmark. Experiments on AGIEval-MATH, GPQA-Diamond, WSC, BBH-Navigate, and StrategyQA with o3-mini, o4-mini, and DeepSeek-R1 report accuracy gains over CoT/SPO and a length-aware baseline (TALE-EP), plus ablations removing Bayesian search or the judge. The paper also presents a case study illustrating when too-short or too-long chains fail and provides qualitative evidence of multi-peak behavior.

**Strengths:**

1. The paper articulates why length control matters and argues (with plots and a case study) that the accuracy–length curve can be multi-peaked, motivating global rather than local search.

2. Method is simple and potentially practical. Using preferential BO with an LLM judge avoids scalar rewards and can be dropped on top of existing prompting schemes (CoT, SPO). The staged procedure and pseudo-code/overview are easy to follow.

3. Across several datasets and models, TBO variants often outperform their base counterparts; ablations suggest both the Bayesian step and the LLM-judge contribute.

4. The worked example convincingly shows failure modes of too-short and too-long chains and how a length around the “sweet spot” can fix them.

**Weaknesses:**

1. Questionable/implausible headline numbers and limited statistical reporting.
Some reported accuracies seem unusually high, which is surprising for this hard benchmark and calls for careful audit, confidence intervals, repeated trials, and per-seed variance.

2. Potential evaluation leakage / selection on test distribution.
The method selects a task-level length using about 1% of the benchmark samples and then evaluates on the full test set. Even if labels aren’t directly used in selection, this is still hyper-parameter tuning on the test distribution and risks optimistic estimates. A clean setup would tune on a disjoint validation split (or per-task train/dev) and report on untouched test.

3. LLM-as-a-judge dependence lacks robustness analysis.
The paper claims LLM judges are more consistent than humans in their setting, but details are light (e.g., judge prompt, calibration, inter-judge agreement metrics, cost/latency). Using GPT-3.5 for filtering adds another moving part. A deeper study (cross-judge agreement, judge ablations, label noise sensitivity) is needed.

4. Methodological inconsistencies in Section 2.2.
The description mixes length selection with a delete-and-replace strategy on fixed-length sequences and bottom-performing segments, which reads like sequence-editing rather than selecting a scalar token budget. This raises clarity concerns about what exactly is optimized each iteration.

5. Task-level, not instance-level adaptation.
The method converges to a single length per task class. Table 3 itself shows wide variance in token needs (e.g., GPQA), where per-instance or per-cluster adaptation would be more appropriate; unsurprisingly, the paper notes reduced gains when difficulty varies. A direct comparison to per-instance length policies (learned or heuristic) is missing.

6. Baselines could be stronger and better contextualized.
TALE-EP and simple CoT/SPO are included, but the paper omits competitive adaptive test-time compute methods (e.g., per-instance early/stop rules, token-elasticity with stronger controllers, or modern step-budget schedulers). Even within its framing, per-task vs. per-instance comparisons are essential to position TBO.

**Questions:**

1. GPQA numbers sanity check. Please provide per-seed means/SDs and re-runs for the unusually high accuracies (e.g., o4-mini SPO+TBO on GPQA). Include per-question breakdowns and auditors’ scripts to confirm scoring.

2. Can you re-run with a proper split (tune on a separate validation subset, report on test) and quantify the performance drop, if any, vs. tuning on 1% of the test? or deeper case-by-case qualtitative anaylsis can make the work robust.

3. Judge robustness. Report inter-judge agreement across 3+ judges (GPT-4-class, o3-class, and a smaller model) and show how accuracy changes when the judge is swapped.

4. Instance-adaptive baselines. Please compare against per-instance controllers (e.g., confidence- or margin-based early-stop/extend, token-elastic methods with per-item adaptivity) using the same judge budget. How far is task-level TBO from instance-level upper bounds?


5. Statistical reporting. Report numbers over multiple seeds for all tables; include n-wise exact match and robustness under noisy judges.

---

### Official Review · Reviewer_7yWD · 2025-11-05

**Soundness:** 2
**Presentation:** 2
**Contribution:** 2
**Rating:** 4
**Confidence:** 2

**Summary:**

This paper introduces Token Bayesian Optimization (TBO), a novel unsupervised framework that automatically identifies the optimal reasoning length for large language models (LLMs). By framing token-length selection as a preferential Bayesian optimization problem guided by LLM-as-a-Judge feedback, TBO adaptively balances reasoning quality and computational efficiency without supervision or task-specific tuning. Experiments across multiple reasoning benchmarks and models demonstrate consistent improvements (up to +20%) and reduced token usage, revealing a previously overlooked multi-modal relationship between reasoning length and performance. The approach is innovative, broadly applicable, and well-validated, though further theoretical grounding and cost analysis would strengthen the work.

**Strengths:**

- High Novelty – This is the first systematic attempt to couple Bayesian Optimization with reasoning-length control in LLMs.
- Task-Agnostic and Unsupervised – TBO requires no labeled supervision or reward modeling, integrating seamlessly with various reasoning strategies (e.g., CoT, SPO).

**Weaknesses:**

- Imprecise Definitions and Assumptions: The concept of a "global optimal reasoning length" is not rigorously defined, making it hard to verify claims empirically. Additionally, the reliance on LLM-as-a-Judge for performance evaluation introduces potential biases and variability, as different judges might yield inconsistent rankings.
- Incomplete Baseline Comparisons: Some competing methods (like LCPO or other RL-based approaches) aren't fully replicated or tuned in the experiments, potentially skewing the reported advantages of TBO. Fairer, head-to-head evaluations with standardized hyperparameters would bolster credibility.
- Limited Generalization Across Tasks: TBO assumes a consistent length-performance landscape, but results (e.g., Table 3) show performance drops in high-variance or mixed-difficulty tasks. The framework's "task-agnostic" claim feels overstated without exploring adaptations for diverse or evolving task distributions.

**Questions:**

See weaknesses

---

### Meta-Review · Area_Chair_cLTz · 2026-01-06

**Summary:**

The submission proposes Token Bayesian Optimization (TBO), a supervision-free procedure to select an appropriate reasoning token budget for LLM prompting (e.g., CoT, SPO). The key motivation is that task accuracy as a function of reasoning length can be non-monotonic and multi-peaked, so fixed budgets or monotone heuristics can be suboptimal. TBO uses preferential Bayesian optimization (GP surrogate) driven by an LLM-as-a-judge that provides listwise rankings of candidate lengths on a small subset of instances, then applies the selected task-level length to the full benchmark.

Across several reasoning benchmarks and three models, reviewers agree the idea is interesting and potentially practical, but they raised substantial concerns about evaluation protocol (possible tuning on test distribution), statistical validity and plausibility of reported numbers, clarity and reproducibility of the method description, and robustness to judge choice and task heterogeneity.

**Reviewer Concerns:**

No author rebuttal or discussion.

**Reviewer Scores:**

All reviewer scores should be unchanged.

---

### Decision · Program_Chairs · 2026-01-26

Reject